



# Atmospheric River Induced Precipitation in California as Simulated by the Regionally Refined Simple Convective Resolving E3SM Atmosphere Model (SCREAM) Version 0

Peter A. Bogenschutz[1], Jishi Zhang[1], Qi Tang[1], and Philip Cameron-Smith[1]

[1]Lawrence Livermore National Laboratory, Livermore, CA

**Correspondence:** Peter Bogenschutz (bogenschutz1@llnl.gov)

**Abstract.** Using the Regionally Refined Mesh (RRM) configuration of the U.S. Department of Energy's Simple Cloud Resolving E3SM Atmosphere Model (SCREAM), we simulate and evaluate four meteorologically distinct atmospheric river events over California. We test five different RRM configurations, each differing in terms of the areal extent of the refined mesh and the resolution (ranging from 800 m to 3.25 km). We find that SCREAM-RRM generally has a good representation of the AR

generated precipitation in CA, even for the control simulation which has a very small 3 km refined patch, and is able to capture the fine scale regional distributions that are controlled largely by the fine scale topography of the state. Although, it is found that SCREAM generally has a wet bias over topography, most prominently over the Sierra Nevada mountain range, with a corresponding dry bias on the lee side. We find that refining the resolution beyond 3 km (specifically 1.6 km and 800 m) has virtually no benefit towards reducing systematic precipitation biases, but that improvements can be found when increasing the

areal extent of the upstream refined mesh. However, these improvements are relatively modest and only realized if the size of the refined mesh is expanded to the scale where employing RRM no longer achieves the substantial cost benefit it was intended for.

## 1   Introduction

A literal interpretation of the cliche "when it rains it pours" could not be more true for California. In a state that has recently been subjected to prolonged periods of drought, California is also prone to excessive precipitation that is capable of producing valley flooding and copious amounts of snowfall at the higher elevations. The vast majority of extreme precipitation events in California are associated with Atmospheric Rivers (ARs), which play a crucial role in delivering rain and snow to the western United States (Ralph et al. 2004; Leung and Qian 2009). While the majority of ARs affecting California are beneficial, they

can also pose significant hazards, primarily in the form of flood risk (Eldardiry et al., 2019), depending on their strength and the timing between individual events. The recent winters of 2016-2017 and 2022-2023 are good examples of AR induced flooding and stressed water supply infrastructures in California (Huang et al. 2018). While both the aforementioned seasons




were characterized by a frequent onslaught of moderate-to-strong ARs, the flooding produced by those events and seasons pales in comparison to historical extremes such as the "Great Flood of 1861-1862" in California. Recent work suggests that the odds of such an event occurring again are increased due to climate change (Huang and Swain 2022).

ARs are characterized by thin streams of moisture originating in the sub-tropics, that are typically associated with extratropical cyclones and are responsible for nearly half of the annual precipitation produced on the West Coast of the United States (Ralph et al. 2019). Today's general circulation models (GCMs) are typically run with horizontal grid sizes on the order of 100 km, to enable the production of multi-decadal future projection simulations using a reasonable amount of computational resources. These standard resolution GCMs are able to sufficiently resolve the large-scale environments and moisture transports associated with AR events and have been used to study how AR strength and frequency may change in a warming climate (Swain et al. 2018, Shields and Kiehl 2016). However, the Western United States (including California) is characterized by complex terrain and topography that plays a crucial role in determining regional precipitation distribution and climatology in which models with a resolution of ∼100 km simply cannot resolve (Caldwell et al. 2019; Delworth et al. 2012; Duffy et al. 2003). In addition to better resolving topography, the simulation of extreme precipitation events improves with higher resolution (Terai et al. 2018; Wehner et al. 2010). It has previously been shown that a minimum horizontal resolution of ∼25 km is needed to produce realistic precipitation patterns in California and the western US (Caldwell et al. 2019; Huang and Ullrich 2017), though recent work suggests that ∼3 km resolution is necessary to capture the key physical characteristics associated with extreme atmospheric river events (Huang et al. 2020; Rhoades et al. 2023).

The requirement for a horizontal resolution of 3 km is obviously much finer than the current 100 km "workhorse" GCMs that are used extensively as of present. However, the next generation of GCMs, taking the form of global storm resolving models (GSRMs), have been developed and the first intercomparison project of such models has already been performed for short duration simulations (Stevens et al. 2019). These GSRMs are usually run with horizontal grid sizes of 1 to 5 km and typically do not include parameterized deep convection, thus they rely on the fluid dynamics rather than parameterized physics to represent deep cumulus convection.

The Simple Cloud-Resolving E3SM model (SCREAM; Caldwell et al. 2021) is a new global convection permitting model (GCPM) developed by the U.S. Department of Energy. With a global resolution of 3 km, Caldwell et al. (2021) shows that SCREAM ameliorates many long standing biases typically associated with conventionally parameterized GCMs, such as the Energy Exascale Earth System Model (E3SM), that are typically run with horizontal resolutions on the order of 100 km. These include the representation of coastal subtropical stratocumulus, the vertical structure of tropical convection, the frequency of light vs. heavy precipitation, and biases associated with Amazonian precipitation. Caldwell et al. (2021) also shows that SCREAM can well represent the climatological AR frequency and strength when compared to observations.

While it is currently exorbitantly expensive to use SCREAM for a multi-decadal simulation, it is possible to leverage the regionally refined mesh (RRM; Ringler et al. 2008) to allow a region of interest to be run at cloud permitting resolutions. The RRM approach has been successfully used over the past decade in conventional GCMs to reap the benefits of local high resolution, but with a fraction of the computational cost compared to running the full model at high resolution (e.g. Zarzycki and Jablonowski 2015; Tang et al. 2019; Tang et al. 2023; Bogenschutz et al. 2022). Liu et al. (2022) recently used the RRM





approach for SCREAM to simulate a strong derecho event over North America. In their approach they refined the horizontal resolution to 1.625 km over their region of interest and 25 km over the rest of the globe. They found that SCREAM had an
excellent representation of this particular event and demonstrated that the RRM approach provides a suitable testbed to evaluate how SCREAM performs for individual extreme weather events. Using the Community Earth System Model (CESM; Hurrell et al. 2013), Rhoades et al. (2020) simulated the U.S. Winter Hydroclimate using various RRM configurations and found that resolution was more important to accurately simulate precipitation than the size of the refined mesh.

The recent work of Zhang et al. (2024) used SCREAM with an RRM over California to produce future projection climate
simulations. That work represented the first time that SCREAM has been employed to produce climate length simulations, thanks to the reduced computational cost provided by RRM. They found that SCREAM produced a much more realistic regional distribution of climatological precipitation for CA when compared to E3SM, though with a fairly substantial positive precipitation bias compared to observations. They hypothesized that this wet bias is due to a combination of large-scale biases in their E3SM forcing data and errors associated with the physical parameterizations of SCREAM.

In this paper we exploit the fidelity of SCREAM to simulate individual AR events over CA by running hindcasts of four meterologically diverse precipitation events initialized from reanalysis. We run an identical RRM configuration that was used in Zhang et al. (2024) and also explore the sensitivity to both the resolution and size of the refined mesh. Can SCREAM simulate individual AR events for CA with fidelity? Does the forecast of precipitation improve when the resolution is increased to 1.6 km and 800 m? Does the size and location of the refined mesh region matter when simulating precipitation for CA? These
are the types of questions we seek to address in this study. In related work, Bogenschutz et al. (2023) found that SCREAM in an idealized doubly-periodic configuration was reasonably scale aware and insensitive for most cloud regimes when run with horizontal resolutions from 800 m to 5 km though, that work did not consider the influences of topography. We recognize that while the role of resolution and refined domain size for the simulation of U.S. west coast landfalling ARs has already received attention (Huang et al. 2020; Rhoades et al. 2020; Rhoades et al. 2023), our work will test finer resolutions and smaller refined
domain sizes than previous studies.

This paper is organized as follows: section 2 gives a description of SCREAM and RRM used in this study, while section 3 provides details for the experiment design including the grid configurations used, the individual AR cases selected, and the initialization and evaluation data sets utilized. Results are presented in section 4 and we conclude with a summary and discussion in section 5.

## 2 Model Description

### 2.1 SCREAM

We use a model version very similar to SCREAMv0, as described in Caldwell et al. (2021). The development of SCREAM is designed to fulfill the US Department of Energy (DOE) mission of focusing on compute-intensive frontiers in climate science. In SCREAM the dynamical core consists of the new nonhydrostatic version of the High Order Method Modeling Environment
(HOMME-NH; Taylor et al. 2020). Most GSRMs use a very simple boundary layer scheme but SCREAM is unique since it





uses the Simplified Higher Order Closure (SHOC; Bogenschutz and Krueger 2013), which is a unified cloud macrophysics, turbulence, and shallow convective parameterization based on a double-Gaussian assumed probability density function. The Predicited Particle Properties (P3) scheme is used for microphysics (Morrison and Millbrant 2015), while gas optical properties and radiative fluxes are computed using the RTE+RRTMGP radiative tansfer package (Pincus et al. 2019). SCREAMv0 used a prescribed-aerosol version of E3SM's modal aerosol model, however in this work we use a version of SCREAM that prescribes both cloud-condensation nuclei number and aerosol radiative properties from an E3SMv2 simulation. This is known as Simple Prescribed Aerosol (SPA) and will be incorporated into SCREAMv1.

## 2.2 Regionally Refined Model

In this study we capitalize on the Regionally Refined Mesh (RRM) capability within the E3SM/SCREAM tool chain (Tang et al. 2019; Tang et al. 2023; Bogenschutz et al. 2022; Zhang et al. 2024). The RRM approach serves as an efficient framework and test bed for the development and analysis of high horizontal resolution models. The simulation cost of RRM is primarily influenced by the high-resolution region, making the model cost proportional to the size of the region with the finer mesh. For instance, a high-resolution mesh covering 10% of the globe would roughly equate to 10% the cost of running the entire globe at the higher resolution. Consequently, the RRM configuration proves to be an appealing feature for a GCPM like SCREAM, offering substantial computational cost savings, especially when focusing on a single scientific region of interest.

## 3 Experiment Design

This section outlines the methods employed in this study, encompassing the design of the various RRM configurations, a description of the specific AR events, and a overview of the initialization and evaluation data employed.

### 3.1 Grid Configurations

In this study we design and test five different RRM configurations. These configurations differ in terms of the size or resolution of the refined patch. Figure 1 displays the control RRM used in this study. We treat this as the control grid because it is the exact RRM used by Zhang et al. (2024) to produce future projection climate simulations over CA. This grid is progressively refined from the outer global resolution of ne32 (corresponding to a resolution of ~100 km at the equator) to the convection-permitting scale for California (ne1024, ~3 km), with a transition zone between them. We refer to this RRM grid as CA-3km or simply as the control grid interchangeably.

### 3.1.1 Refined Domain Sensitivity

Rhoades et al. (2020) explored the role of the refinement mesh size in CESM and found that resolution over the western U.S. generally played a more important role for accurate simulation of climatological precipitation than the distance of the upstream high resolution mesh from the area of interest. However, the size of the refined mesh in our control CA-3km grid is far smaller than any refined mesh size tested in Rhoades et al. (2020). The CA-3km grid was designed to encompass the entire state of





CA but was intentionally constructed to make the high resolution region as small as possible to mitigate computational cost for the long duration simulations. Recognizing that the incoming meteorology to CA could be impacted due to the closeness of the transition region to the state, we construct two grids to explore the potential sensitivity of having a very small RRM. The first we refer to as EPAC-3km (left panel of Fig. 2). Similar to the control CA-3km grid, the resolution of the refined patch is ne1024 while the resolution of the outer global domain is ne32. However, the refined patch has been substantially expanded west into the eastern Pacific Ocean. This allows for the transition region to be further away from the area of interest (CA) and is similar to other nested regional models used for CA AR simulations (e.g. Huang et al. 2020). The upstream refined mesh distance from CA for EPAC-3km is also similar to the smallest refined domain used in Rhoades et al. (2020) for CESM.

The second grid designed to test the domain sensitivity is referred to as GLBP-3km (right panel of Fig 2). This grid has the same resolution specifications in the refined and global regions when compared to the CA-3km and EPAC-3km configurations but the refined region has been substantially expanded to the north, south, and west; while modestly expanded to the east. We consider (and will occasionally refer to) this domain as our "global proxy" and serves as a substitute to running our cases using global 3 km SCREAM to mitigate computational expense. The GLBP-3km grid was deliberately constructed so that all incoming meteorology (i.e. atmospheric rivers and associated storm systems) to CA would be contained within the high resolution mesh for the entire simulation duration for all cases

Table 1 summarizes the specifics of the grids used, including the number of columns and the time steps for each configuration. We note that the HOMME dycore consists of spectral elements, with each element containing $4 \times 4$ grid of Gauss-Lobatto-Legendre (GLL) nodes, while the physics is handled by a uniformly spaced $2 \times 2$ grid (called "pg2" grid), which substantially increases the model throughput (Hannah et al. 2021).

### 3.1.2 Refined Resolution Sensitivity

To test the resolution sensitivity of SCREAM in simulating CA ARs we construct two grids with higher resolution within the refined patch. For both these grids, the areal extent of the refined patch is exactly the same as that in the CA-3km control grid (Fig. 1). However, in one we increase the resolution of the refined patch to ne2048, corresponding roughly to a resolution of 1.6 km, and we refer to this grid as CA-1.6km. For the other grid we increase the resolution of the refined patch to ne4096, corresponding to a resolution of approximately 800 m, and this grid is referred to as CA-800m. Table 1 provides details for the various grid configurations used in this study.

### 3.2 Atmospheric River Event Cases

To assess the skill of SCREAM RRM in simulating Atmospheric Rivers (ARs), we carefully choose four meteorologically distinct events. It's important to clarify that while some of these events may have had significant societal impacts, our selection criteria aimed to assemble a diverse set with varying meteorological setups and regional effects, rather than focusing solely on the most costly or disruptive occurrences. Our chosen cases include two events predominantly affecting northern California, one centered on central California, and another making landfall in Southern California. The following section provides a brief introduction for each case, along with the rationale behind their selection.





### 3.2.1 October 2021 - Northern California (NC2021)

On October 24, 2021 an "exceptional" category 5 (Ralph et al. 2019) AR effected primarily northern CA. This AR was associated with a parent low that underwent rapid intensification in the northeast Pacific. While the center of this low made landfall well north of CA, this system was associated with a potent and long stream of moisture tapped into the sub-tropics that was aimed directly at CA. This storm delivered unseasonable and exceptional precipitation accumulations (as high as 375 mm or 14 inches) to northern California and the Sierra Nevada, in addition to damaging winds in excess of 27 m/s at most locations

that resulted in widespread power outages. Two very active fire seasons experienced in CA prior to this event contributed to debris/mud flows and flooding in areas of burn scars. Although the higher elevations in the Sierra Nevada experienced a noteworthy amount of snowfall for October, this was predominantly a warm atmospheric river event, with the vast majority of precipitation falling as rain rather than snow over the higher terrain. Figure 3a displays the ERA5 integrated vapor transport (IVT) during peak intensity at landfall on October 25, 2021 at 00:00 UTC.

This marked an unusually potent atmospheric river impacting California, an occurrence that typically transpires once every five to ten years (Ralph et al. 2019). Particularly noteworthy is the timing of this event in October, a period deemed as California's "shoulder season" for precipitation. However, despite the strength of this storm, the hydrologic impacts produced were relatively muted due to the dry antecedent conditions present in state. In addition to being the first major storm of the season, CA was experiencing its most intense drought in recorded history with nearly 60% of the state classified in the D4 or

"exceptional drought" classification. Had this AR occurred in the middle of an active period, such as the recent storm cycles of 2016-2017 or 2022-2023, with a pre-existing snowpack, saturated soils, and reservoirs at maximum capacity; the impacts of this storm would have likely been much more severe (such as in the 1997 CA flood event; Lott et al. 1997). Therefore, it is important to evaluate SCREAM's ability in representing this type of strong and warm AR.

### 3.2.2 January 2017 - Northern California (NC2017)

The 2016-2017 water year was truly exceptional, being one of the wettest recorded for Northern California and helping the state escape its longest drought in recorded history. January 2017 was one of the most active periods during this year, responsible for delivering a parade of moderately strong ARs in rapid succession. This series of storms resulted in overflowing rivers, saturated soils, and increased risk of mudslides in areas affected by wildfires in previous years. The Oroville Dam, the tallest dam in the United States, faced a critical situation as its spillway suffered damage, prompting evacuations downstream due to

the potential for catastrophic failure. In addition, unlike the NC2021 event, this series of storms produced copious amounts of Sierra snowfall which led to widespread travel disruptions and temporary closures of many ski resorts.

In this study, our focus centers on simulating a five-day time frame spanning from January 7th to the 11th, during which two noteworthy atmospheric rivers (ARs) unfolded. The initial AR, which manifested on January 8th, proved to be the more formidable of the two, marked by thunderstorms featuring frequent hail and lightning, the highest cumulative rainfall and

185 snowfall, and the most substantial hydrological impacts. Classified as a category 4 event, it was promptly succeeded by a category 3 AR on January 10-11. This specific period is chosen to leverage SCREAM's capability in modeling consecutive





and robust AR occurrences, coupled with the substantial snowfall accumulation in the Sierra Nevada. Figure 3b provides a depiction of the IVT during the first AR event at January 8, 2018 18:00 UTC.

### 3.2.3    January 2021 - Central California (CC2021)

Opposite the 2016-2017 water year, the 2020-2021 water year was among the driest ever recorded for several regions of the state. While the storm track was active this year, the large-scale pattern was influenced by a strong La Niña with a Pacific high pressure ridge that directed the vast majority of storms towards the Pacific Northwest and only bringing California light showers. However, this year did produce one notable storm that was characterized as a cold AR associated with a strong cold front that stalled over the central coast of CA for several days. Localized regions of the Santa Lucia coastal mountain range received upwards of 450 mm (18 inches) of rain for this event and washed out sections of the famous Highway 1 in Big Sur. This storm was also notable for not only producing an abundance of mountain snow in the Sierra Nevada but also the relatively rare snow accumulations in the hills of San Francisco Bay Area and Transverse Ranges in southern CA.

Preceding this storm, a ridge of high pressure was sitting over the southeastern Pacific. The anti-cyclonic rotation of the high pressure pushed a large swath of moisture northward toward Alaska. This moisture was directed around the central high pressure and then southward toward CA on January 26-29, 2021. Therefore this storm was characterized as a "cold" AR, which is fairly atypical as CA is generally more susceptible to "warm" ARs that have a direct connection to moisture from Hawaii or the sub-tropics. While this storm was not unusually strong (especially when compared to the NC2021 or NC2017 cases), ranking as a category 3 AR, this represents an ideal case study to assess SCREAM's performance in simulating not only an AR with cold origins but also one that stalled over the complex terrain of Big Sur for several days. The IVT as seen by ERA5 on January 28, 2021 00:00 UTC as it stalled along the central coast can be seen in Fig. 3c.

### 3.2.4    January 2005 - Southern California (SC2005)

Late December 2004 through early January 2005 was a profoundly wet period for all of California, but this was especially true for the southern reaches of the state. The January 7-11 time period was particularly wet for southern California with precipitation that continued nearly unabated and caused 10 deaths due to a landslide into homes in a coastal town in Ventura County. This storm was also responsible for causing millions of dollars of flood and storm-related damage throughout the region and thus is probably the most impactful storm included in this study. Isolated areas in the Transverse Mountain ranges received upwards of 750 mm (30 inches) of precipitation, with widespread reports of accumulations over 250 mm (10 inches) observed through this range.

The large scale pattern was characterized by a blocking ridge near the Alaskan Aleutian Islands and a quasi-stationary low pressure system off the west coast of the United States, which is an ideal synoptic set up to usher storms into California over a prolonged duration. The most crucial ingredient, however, was the intrusion of subtropical moisture delivered via a moist subtropical jet stream that was aimed directly at southern California. Though this AR only briefly achieved a maximum intensity representative of category 3 status, it was the duration of this AR that was responsible for the most significant impacts over southern California. Figure 3d displays the IVT as depicted by ERA5 on January 10, 2005 00:00 UTC. In addition, the



cold shortwave troughs hitting California from the north and interacting with subtropical moisture helped to produce 7 to 10 feet of snow in the Sierra Nevada. We include this case because of its impact on the Transverse Mountain range and to evaluate how SCREAM simulates long duration stalled AR events.

### 3.3    Initialization and Nudging

All cases are initialized using the fifth generation of the European Center for Medium Range Forecast Reanalysis (ERA5; Hersbach et al. 2020). Each simulation is initialized with reanalysis at approximately 24 hours before the AR makes landfall
on CA. The initialization time and the simulation length for each case are noted in Table 2. We do note that sensitivity tests were conducted for select cases to assess the skill when increased/decreased forecast lead time is applied and those results will be noted in the results section. In addition, unless otherwise stated, results for all cases and grid configurations presented are from a single realization. In section 4.4 we discuss the sensitivity of our results when ensemble members are considered.

All results presented in this paper are from "free running" simulations, meaning that no nudging to ERA5 is applied after initialization. SCREAM RRM functionality allows for nudging to be applied in select regions of the globe, enabling us to nudge the coarse outer domain while letting the high resolution mesh to be freely simulated. When evaluating the initial performance of SCREAM RRM in simulating each case, we did perform experiments using the CA-3km grid where nudging to ERA5 was applied in the coarse domain. The results of these experiments differed little when compared to the free running
simulations. Therefore, we decided to only present results of the free running simulations to eliminate any uncertainty that might be introduced in our nudging strategy (i.e. sensitivity to nudging time scales, etc.).

### 3.4    Evaluation Data

To conduct a thorough assessment of atmospheric river (AR) events in California, it is crucial to compare them against observational datasets with ample temporal and horizontal resolution. This strategy is important as the expected advantages of
high-resolution simulations are more likely to become apparent on smaller temporal and spatial scales. To evaluate the simulation of precipitation and temperature, we use the 4 km PRISM (Parameter-elevation Regressions on Independent Slopes Model) dataset (PRISM Climate Group, Oregon State University, https://prism.oregonstate.edu, accessed 16 Dec 2020). The PRISM dataset adopts the philosophy that "elevation is the most important factor in the distribution of climate variables" for a localized region. For example, PRISM utilizes regression functions correlating precipitation and elevation within specific cat-
egories, distinguished by slope orientation. This differentiation enables a precise evaluation of precipitation on both windward and leeward slopes.

To assess the snow water equivalent (SWE), we employ assimilated snow observations provided by the University of Arizona (UA). The UA SWE dataset (Zeng et al. 2018), sourced from in-situ measurements within the Snow Telemetry (SNOTEL) network and Cooperative Observer Program, integrates temperature and precipitation data from PRISM. This extensive 40-
year dataset boasts a spatial resolution of 4 km. To evaluate large-scale variables, such as integrated vapor transport (IVT), we use the ERA5 global reanalysis (Hersbach et al. 2020). In addition to the observation-based gridded UA product we also compare select points to the SNOTEL network (https://nwcc-apps.sc.egov.usda.gov/imap/, last access: 15 February 2024). We



choose four sites to compare to (Table 3): Independence Lake, Tahoe City Cross, Ebbetts Pass, and Virginia Lakes. All four sites are found in the Sierra Nevada Mountain range; the first two reside fairly close to each other, straddling the I-80 corridor,
but differ by more than 450 m (1500 ft) in elevation. Ebbetts Pass and Virginia Lakes sites are both found in the central Sierra Nevada; with the former located at pass level and the later located on the lee-side of the mountain range. Though the Ebbetts Pass site is located at pass level, the Virginia Lakes site resides on a ridge and is actually at higher elevation compared to Ebbetts Pass.

## 4 Results

We primarily evaluate the proficiency of the SCREAM RRM in simulating California atmospheric rivers (ARs) by emphasizing its capability to replicate observed patterns of accumulated precipitation and snowfall. Additionally, we examine how well the model captures the structure and magnitude of Integrated Vapor Transport (IVT) throughout the duration of the events. Figure 3 illustrates the Integrated Vapor Transport (IVT) during peak intensity as observed in the ERA5 reanalysis for each case. Conversely, Fig. 4 presents the corresponding information for SCREAM simulations conducted on the CA-3km grid. The
analysis of the two figures reveals that, overall, SCREAM effectively reproduces the magnitude and structure of IVT compared to ERA5 reanalysis, albeit with some discernible discrepancies. In the NC2017 case, SCREAM exhibits a slightly heightened IVT magnitude offshore but weakened magnitudes inland. Conversely, in the SC2005 case, the SCREAM-simulated IVT appears somewhat narrower and more intense, in contrast to the broader and relatively weaker magnitudes observed in the ERA5 IVT. Examining the IVT structure at one simulated time, however, is quite limiting and a more comprehensive analysis
of the simulated IVT will be examined in greater detail. This comparison is presented to demonstrate the general characteristics of the simulated IVT by SCREAM RRM on the CA-3km control grid.

### 4.1 Accumulated Precipitation

The event accumulated precipitation skill scores, comprised of the root mean squared error (RMSE), bias, and correlation coefficient for each case and for all grid configurations are displayed in Fig. 5. For all scores and cases, PRISM is used as
the observational data source. One key takeaway from this figure is the proficiency of SCREAM in representing the warm atmospheric river (AR) of NC2021. While it is apparent that SCREAM suffers from a statewide positive precipitation bias for all cases and grid configurations, this bias is muted for the NC2021 case. Conversely, SCREAM has the least skill in representing the relatively colder (and successive) AR events that occurred in the NC2017 case.

An observation evident from Fig. 5 is that resolution seems to have minimal impact on precipitation skill. For instance, the
CA-1.6km (ne2048) and CA-800m (ne4096) grids do not demonstrate improved skill scores when compared to the CA-3km grid. Though it should be noted that to compute the skill scores all grids were remapped to the PRISM grid; which is a bit lower resolution than the CA-1.6km and CA-800m grids and there can be implications when interpreting these skill scores when the model resolution goes beyond the observations (Risser et al. 2019).





Perhaps the most interesting piece of information inferred from Fig. 5 is that most grid configurations generally cluster
around one another; with a couple notable exceptions. Chiefly, for the CC2021 cold AR case and the NC2017 case we note that
the GLBP-3km grid (which serves as our global high resolution proxy) exhibits comparatively greater skill. Specifically, most
grid configurations tend to have the lowest pattern correlation scores for the CC2021 case, however, the "global proxy" grid is
an outlier. Additionally, in the NC2017 case we see that the GLBP-3km grid has noticeably smaller RMSE and bias scores.

To gain a deeper understanding on the characteristics of regional precipitation distributions in SCREAM RRM, we examine
the observed versus simulated event accumulated precipitation in Fig. 6. Since we found that most SCREAM RRM grid
configurations tend to cluster around one another, we display three select grids; specifically the CA-3km, CA-800m, and
GLBP-3km grids. While Fig. 6 displays the event accumulated precipitation, this figure is complemented by Fig. 7 which
displays the regional biases for each grid configuration. We refer to both figures in tandem for the following analysis, in which
we will systematically discuss results for each individual case.

The top row in both Figs. 6 and 7 confirm the satisfactory and robust performance of SCREAM RRM in the simulation
of the category 5 warm NC2021 case. Evident in all three grid configurations shown is a good representation of the regional
precipitation distributions, with a maximum along the northern Sierra Nevada mountain range, secondary maximum along
the north coastal ranges, and minima in the central valley. The relatively more modest precipitation totals in central and
southern California are well represented in addition to precipitation maxima of the smaller mountain ranges, such as the Santa
Cruz mountains. Fig. 7 (top row) reveals that all configurations have a positive precipitation bias in areas of topography (most
conspicuous in the Sierra Nevada range) and dry bias in valleys and areas downwind of topography (i.e. the northern CA central
valley and western Nevada) for this case. Though the GLBP-3km simulation has a statewide mean precipitation accumulation
that agrees best with observations, it is evident from Fig. 7 that this is largely due to compensation of a stronger central valley
dry bias when compared to the other grid configurations.

Regional precipitation distributions and biases for the NC2017 case are displayed on the second row of figures 6 and 7,
respectively. As already discussed, this is the case where all SCREAM RRM configurations tend to deliver the poorest pre-
cipitation skill scores. This is made evident by the large precipitation biases, though as already pointed out, these biases are
muted for the GLBP-3km grid. The majority of the positive precipitation bias in the CA-3km and CA-800m cases is centered
on the Sierra Nevada, with large biases also seen in coastal Central California (i.e the San Francisco east Bay Area and Big Sur
region). In the GLBP-3km grid the Sierra Nevada bias is still prominent, yet substantially reduced, while the bias in the east
Bay Area and Big Sur regions are largely eliminated; both of which are responsible for the improved skill scores for this grid.
However, it should be noted that all grid configurations perform well for the northern coastal range areas.

Unlike the previous two cases in which northern California was the focus of the landfalling ARs, the CC2021 represents a
case in which a strong AR stalled over central California (third row of figs 6 and 7). Also unlike the NC2021 and NC2017
cases, most grid configurations suffered from lower pattern correlation scores. This is evident when examining the regional
precipitation distributions. PRISM observations shows well defined maxima over the Santa Lucia coastal mountains in Central
CA and the southern Sierra Nevada mountains. While all SCREAM RRM configurations represent the spatial distribution of
the Sierra maxima (albeit with a positive bias), the coastal maxima is displaced to the south over the Transverse Mountain





range in the CA-3km and CA-800m grids. While the GLBP-3km grid also has a wet bias over the Transverse Mountain range,
the placement of the geographically small maxima along the central coast is well represented. This is another interesting
difference/improvement pertaining to the GLBP-3km grid that will also be discussed in further detail.

The bottom row of Figs. 6 and 7 displays the accumulated precipitation and associated biases, respectively, for the long
duration AR event that was focused on southern CA. PRISM observations shows observed maximum precipitation along the
Transverse range in Ventura County, with another maximum associated with the more inland ranges (i.e. San Gabriel and San
Jacinto mountains). While all configurations represent the maximum of the inland mountain ranges well, all fail to capture
the absolute maximum in the Transverse Range, as evidenced by localized dry bias larger than 250 mm (which represents the
worst local bias for all of the cases included in the study). This isn't too surprising when comparing the simulated IVT to that
of reanalysis (Figs. 3 and 4), which showed that SCREAM's moisture plume was too narrow for this case. It is interesting that
neither resolution nor refined domain size improves this representation, and it is worthwhile to note that Huang et al. (2020)
simulated this same case using WRF which also had a substantial dry bias in the Transverse Range. While the Sierra Nevada
received significant precipitation during this event, that precipitation was largely attributed to several passing shortwaves during
this time period aided by weak AR moisture. However, it is interesting to note that the Sierra precipitation bias for this case is
far muted when compared to the CC2021 and NC2017 cases.

While each case certainly contains their own unique regional precipitation characteristics and model biases, some common
themes and connections to other work can be made. First, it is apparent that SCREAM tends to overestimate precipitation
relating to topography that is fueled by a large-scale moisture source. For all cases examined, there is a modest-to-strong
precipitation bias over both the major and minor mountain ranges of CA. This is complimented by a prominent dry bias
downwind of topography and is most apparent along the CA/NV border east of the Sierra Nevada. Another common theme
is the positive statewide precipitation bias, which ranges from 10 to 35% depending on the case. This is in comparison to the
climatological precipitation bias produced by the CA-3km grid (Zhang et al. 2024) of 129%. Zhang et al. (2024) hypothesized
that the strong precipitation bias in their simulations was attributed to a combination of errors in the large scale forcing (which
was nudged to a 100-km E3SMv1 run in the coarse outer domain) and biases in the SCREAM physics. The smaller biases
produced in our hindcast simulations suggest that a majority of the bias produced by SCREAM RRM climate runs likely stems
from the former; with relatively smaller contributions by the later. GCMs typically underestimate the strength and duration of
high pressure blocking ridges that dominate the dry years in California (Davini and Dandrea 2020; Schiemann et al. 2020),
which likely led to higher than observed frequency of storms in the CA RRM downscaled simulations performed in Zhang et
al. (2024).

Further, it is interesting to note the lack of importance resolution plays in SCREAM RRM AR simulations. In a sense,
it is a positive result that SCREAM is not dominated by undue resolution sensitivity and is consistent with the findings of
Bogenschutz et al. (2023) which found SCREAM to be insensitive to resolution for grid boxes sizes ranging from 800 to 5
km in a doubly periodic configuration. It wasn't until resolution approached that of large eddy simulation (LES) that they
noted a sensitivity, but this was only noticeable in the shallow convective regime. However, their simulations did not include
topography. While our simulations show that precipitation can be better resolved in areas of fine scale topography, there



is no discernible benefit in terms of skill scores and bias reduction when moving to 800 m resolution. Finally, it appears
that increasing the area of the refined mesh yields improvements over the CA-3km grid, but this is only true if the mesh is
substantially refined beyond what is typical for a RRM. We note that these results (pertaining to the importance of resolution
vs. refined domain size) are opposite of the conclusions found in Rhoades et al. (2020), who ran various grid configurations
with a 25 km refined mesh for the Western United States. However, we test grids with smaller refined mesh sizes and with
much higher resolution. Combining our findings with those of Rhoades et al. (2020) and Huang et al. (2020) indicates that
prioritizing resolution is crucial, until approximately 3 km is attained. Beyond this point, enhancing the upstream refined mesh
size may yield some incremental improvements.

## 4.2 Accumulated Snowfall and Temperature

The snowpack plays a crucial role in California's water resource management. Given the distinct intermittence of precipitation
in the state, marked by dry summers and comparatively wet winters, the mountain snowpack serves as California's "frozen
reservoir". This essential reservoir aids in generating runoff and mitigating wildfire risk during the dry season (Bales et al.
2011). Therefore, in order to adequately assess changes in snowpack to greenhouse gas emissions, a climate model should be
able to simulate the CA snowpack with fidelity. Zhang et al. (2024) found that the CA RRM produced superior skill towards
simulating the snowpack when compared to the 100 km E3SM. The latter, being too coarse, failed to capture discernable snow
patterns essential for assessing climate risks. While Zhang et al. (2024) found that the CA RRM overestimated the climato-
logical snowpack when compared to observations, this is not surprising given the tendency for their model to overestimate
precipitation. Here we evaluate SCREAM's ability to simulate the accumulated snow for our AR events.

Figure 8 displays the skill scores for all of our grid configurations for the event accumulated water equivalent snow depth.
To compute these scores we use the University of Arizona snow water equivalent (SWE) as described in Zeng et al. (2018).
While it is unsurprising, we note that the bias and RMSE scores for SWE closely mirror those for the total precipitation rate.
While SCREAM RRM has nearly zero bias for the warm AR NC2021 case, relatively larger biases are seen for the remaining
cases. Though, it is worthwhile to note that the GLBP-3km grid has a very low bias for the NC2017 case for which all other
grid configurations overestimate. In fact, similar to the scores of total precipitation (Fig. 5), the GLBP-3km grid tends to score
best (albeit by a modest amount for most cases) compared to the rest of the grid configurations. In general, all 3 km (ne1024)
configurations score similar for all cases, in terms of correlation, while the higher resolutions have somewhat less skill; which
could stem from capturing smaller-scale variations in the snowfall distributions that may not be present in relatively coarser
resolution dataset (Lundquist et al. 2019).

The regional distributions of the event accumulated SWE are displayed in Figure 9. Similar to the precipitation distributions,
we only display the observations, CA-3km, CA-800m, and GLBP-3km grid configurations. Here we focus on some key general
points; the first is that for all cases each grid configuration adequately represents the regional distribution of snow in the Sierra
Nevada mountain range, though generally with a positive bias. The 3 km and 800 m resolution simulations are able to capture
precipitation maximum/minimum associated with individual ridges and valleys within the Sierra Nevada, though we note that
this distinction is often overdone within the model when compared to the observations. Once again, this provides evidence on



the models tendency to overestimate precipitation in areas of topography. This is further illustrated by examining the snowfall amounts on the leeward side of the Sierra Nevada, where all cases and configurations have a negative bias in far western Nevada.

Figure 9 also highlights the ability of SCREAM RRM, at all resolutions and configurations, to simulate the fine scale details relating to topography. This is most apparent when viewing the snowfall amounts in the relatively lower elevation mountains of the Cascades in Northern California. Additionally, for the cold AR associated with the CC2021 case, we see that all configurations are able to capture the very small areal extent of the low elevation snow fall in the Santa Lucia mountains (Central California, Big Sur region) and the Transverse Mountain ranges in southern California. The proficiency of SCREAM RRM in accurately capturing intricate details bodes positively not just for simulating climate impacts specific to California but also for its broader applicability in global SCREAM simulations. It is also interesting to note that while some of the fine scale details are resolved more sharply in the 800 m (ne4096) case, there is no obvious benefit seen in any cases to using a higher resolution grid in terms of bias reduction. Though again, we note that these high resolution grids are being compared to observational data sets at a coarser resolution. We will discuss the resolution implications further in the discussion section.

A comparison of all grid configurations to SNOTEL observations for each case at select sites can be found in Fig. 3. The Independence Lake and Tahoe City Cross sites both reside in the northern Sierra Nevada and fairly close to one another, though differ by 450 m (1500 ft) elevation. The Independence Lakes site generally receives greater precipitation and snow amounts than Tahoe City and this is true for the SNOTEL observation for all cases. Each grid configuration can capture the distinction between these two sites well; though all seem to underestimate the SWE for the NC2021 case at Independence Lake. Both the Ebbetts Pass and Virginia Lakes sites are located in the central Sierra with the former at pass level and the later on a ridge on the lee side of the mountain range. Therefore, Virginia Lakes tends to receive less precipitation relative to Ebbetts pass, despite being at higher elevation. This is reflected in the SNOTEL observations for each case and generally this distinction is well captured by the model configurations; the exception being the NC2017 case for the CA-1.6km and CA-800m simulations which greatly overestimate the SWE compared to SNOTEL. This is important to note because in general there appears to be no obvious advantages seen in the higher resolution runs for other cases/locations, even when comparing to site specific observations.

Figure 11 displays the skill scores for the averaged daily 2 m temperature, computed relative to PRISM. In general we see that all grids and configurations have high correlation scores, demonstrating the model's ability in representing temperature patterns across the complex terrain of California (as demonstrated in Zhang et al. 2024). With exception of the NC2021 case, all experiment configurations and cases exhibit a positive temperature bias. This positive temperature bias is consistent with Caldwell et al. (2021) who showed that SCREAM suffers from global average 0.7 K positive temperature bias, which is most prevalent over land. It is interesting that the only case which does not have a positive temperature bias (NC2021) also does not have a positive precipitation bias.



## 4.3 Large Scale Moisture Flux

Though high resolution is important to capture the topographic effects and accurate regional distributions of precipitation (Huang et al. 2020), a high fidelity simulation of the synoptic scale IVT associated with ARs is also a crucial ingredient. When constructing the CA-3km grid for long term climate projections, concerns were raised at both the abruptness and closeness of the grid transition to CA. Though 100 km is sufficient to resolve the IVT associated with ARs and Bogenschutz et al. (2023) shows that SCREAM is a reasonably scale aware and insensitive model for idealized cases; it is worthwhile to determine the fidelity of the simulation of the IVT as the grid transitions from 100 km to 3 km (and 800 m) compared to reanalysis and the GLBP-3km global high resolution proxy configuration.

Figures 12 and 13 display the regional distributions of the event maximum and mean IVTs for all events for ERA5 and the selected grid configurations. Among the general conclusions gleaned from these two figures is that all configurations have satisfactory representation of the mean and maximum IVT with reasonable regional distributions. Though it should be noted that all SCREAM simulations are much higher resolution over CA when compared to ERA5, one bias that is obvious is the underrepresentation of the maximum IVT on the leeward side of the Sierra Nevada mountain range. This is consistent with analysis of precipitation and snowfall, showing a strong bias of SCREAM to intercept excessive moisture in areas of topography.

Though the EPAC-3km and CA-1.6km simulations are not displayed, their results and characteristics are very similar to the CA-3km simulation. One outlier is the GLBP-3km configuration which tends to produce lower values of the maximum and mean IVT when compared to ERA5, which is most apparent when examining the NC2017 case. This is interesting because NC2017 was the case where GLBP-3km had improved skill scores, especially pertaining to the reduction in bias, when compared to the other simulations. This could suggest that the improved bias reduction of the GLBP-3km simulation could stem merely from compensating errors. When examining the SC2005 case, it is clear that all grid configurations underestimate the event mean IVT at Point Conception (the prominent "kink" in the CA coastline at approximately 34°N; bottom row of Fig. 13). This is consistent with previous analysis which showed that all grid configurations were unable to represent the absolute maximum precipitation observed on the Transverse ranges for this case; due to the fact that all configurations simulate an IVT plume that is more narrow than observed (as seen in Fig. 4d for CA-3km control configuration). Nonetheless, despite some regional deficiencies, it is encouraging to note that the structure and magnitude of the simulated IVT are fairly robust among the different configurations which have varying resolutions and refinement locations; with no obvious imprints of the grid transitions present.

While figures 12 and 13 provide no information pertaining to the timing and propagation of the IVT plumes through CA, we investigated that there is virtually no difference in this regard among nearly all grid configurations and cases. The one exception is the CC2021 event where the AR stalled over the central California coast (near Big Sur) for over a day. While all grid configurations feature a stalled AR for this case, this tends to occur a bit further south than observed in most grid configurations over the Transverse Mountain Range. This is reflected in the shifted precipitation maximum in most grids compared to observations and relatively poor pattern correlation scores. The exception is the GLBP-3km grid, which does





successfully stall the AR in the correct location. The chief reason for this stall was a region of high pressure near the southern
tip of Baja California; in which a subtle southward error placement of this ridge due to the coarse resolution of 100 km grids in
the outer domain helped to drive the IVT maximum further south than observed. The GLBP-3km grid, on the the other hand,
is able to better resolve the large-scale pattern and hence helps to predict the correct placement of the stalled AR. This is an
example of how high resolution global models can be advantageous over nested/RRM models, as small errors in the mean flow
often translate to local errors in the region of interest.

## 4.4 Sensitivity To Initial Condition

Thus far all results presented have been from a single realization for each case and grid configuration. To help understand
sensitivity to likely errors within the initial conditions we conducted a small 10 member ensemble for all cases for the CA-3km
grid where random perturbations were applied to the initial temperature profiles at all levels and columns. For all cases the
ensemble mean total accumulated precipitation patterns and skill scores are virtually indistinguishable when compared to the
single realization control run (not shown). Figure 14 displays the regional distributions of the event accumulated precipitation
standard deviation for each case for the CA-3km grid ensemble. While localized precipitation amounts exhibit some modest
uncertainty, precipitation patterns are robust.

While not displayed, a small five member ensemble was performed for the GLBP-3km configuration for the CC2021 and
NC2017 cases. These were the two cases in which the GLBP-3km grid demonstrated a clear advantage over the other config-
urations; thus a small ensemble was performed to gain a semblance of confidence that these results were not due to random
chance. Indeed, we found that the relatively high correlation score obtained for the CC2021 case and reduced bias in the
NC2017 case were maintained in the small ensemble runs.

Finally, all experiments performed in this paper were initialized 24 hours before the AR made landfall on CA. To test
the sensitivity to lead time we initialized each case and configuration to be initialized 6 hours before and after the default
initialization time. With exception of the CC2021 case, changing the initialization time did not modify the main findings of this
study. For the CC2021 case we found that initializing 6 hours later led to improved correlation scores for all grid configurations
(with exception of the GLBP-3km case where the score was maintained), due to a more accurate placement of the precipitation
maximum on the central coast rather than the Transverse Range.

## 5 Discussion and Summary

This study uses the Simple Cloud-Resolving E3SM model (SCREAM) and its application in a Regionally Refined Mesh (RRM)
configuration to simulate AR events. The research aims to assess SCREAM's ability to simulate ARs over California, examine
the impact of resolution and RRM fine mesh size, and address questions regarding the fidelity of AR simulations, their im-
proved forecast potential with higher resolution, and the significance of RRM location. The work utilizes four meteorologically
distinct AR event cases and evaluates model performance against observational datasets. This approach aims to enhance our
understanding of AR precipitation representation in not only SCREAM, but climate and weather models at large.



SCREAM effectively reproduces the magnitude and structure of AR-associated Integrated Vapor Transport (IVT) but exhibits some discrepancies. It performs well in simulating warm ARs like NC2021 but has a statewide positive precipitation bias for all cases, with challenges in representing colder ARs, particularly NC2017. Interestingly, resolution doesn't significantly impact precipitation or snowfall simulation. The GLBP-3km global high resolution proxy grid configuration shows

advantages in some cases. SCREAM generally exhibits a positive temperature bias and tends to overestimate precipitation in topographic areas. While SCREAM captures fine-scale details and moisture flux adequately, it underrepresents maximum IVT on the leeward side of the Sierra Nevada. Ensembles and sensitivity tests confirm robustness in the model's results. In conclusion, SCREAM shows reasonable skill in simulating ARs, but addressing biases and improving accuracy in representing AR events and their impacts stands some improvement.

The aforementioned positive precipitation bias seen in our simulations of 10 to 33%, is far less than the the climatological bias produced by the CA-3km grid in Zhang et al. (2024). This suggests that a large majority of the bias produced in that work originates in errors pertaining to the large-scale forcing; with smaller, though non-negligible, errors specific to the SCREAM model. While this work did not address the source of the SCREAM bias, we note that using a framework such as CA-3km hindcasts to perform perturbed parameter experiments would be a relatively efficient way to identify if this bias is related to

the uncertain tunable parameters in the model. Such an exercise would be helpful to improve the topographic precipitation bias that likely exists in regions outside of CA in SCREAM and thus would have positive global impacts on the model simulation.

A key finding of this work is the relative importance of resolution versus refined mesh size. Interestingly, we found no advantage when increasing the resolution from 3 km (ne1024) to 800 m (ne4096) in terms of the precipitation and snowfall skill scores related to ARs. We view this as a positive result in terms of the resolution fidelity of SCREAM and is consistent

with SCREAM's resolution sensitivity of cloud-processes in a doubly-periodic configuration (Bogenschutz et al. 2023). The lack of resolution sensitivity for topographically enhanced precipitation further demonstrates that SCREAM does not possess extreme resolution sensitivity in AR simulations that is driven by either discretization error or model physics sensitivity. While this robust performance may not be generalizable to all models, we conclude that no significant benefits can be gained by increasing resolution within the kilometer scale for AR events if the model is reasonably scale aware and insensitive. We

hypothesize that benefits could potentially be realized should the resolution be increased beyond 400 m, as this is the scale where boundary layer turbulence is permitted and convection resolved (rather than permitted).

Unlike resolution, we did find that expanding the area of the refined mesh produced better skill scores in precipitation and snow. However, this was only achieved when the size of the mesh was expanded significantly to cover the entirety of the east Pacific Ocean. We found that a modest increase to the size of the refined mesh (similar to previous works of nested models when

simulating AR events, i.e. the EPAC-3km configuration) did not produce such benefits, which are only realized if the incoming meteorology to CA is contained with the high resolution region for the entire simulation. While improved precipitation skill scores were achieved with the GLBP-3km grid, the overall characteristics of simulated precipitation produced by that grid was similar to that of the other configurations. In addition, having a refined region as large as the one in the GLBP-3km configuration is counter to the cost-saving strategy that motivates using RRM in the first place. Thus, the CA-3km configuration represents an

appropriate balance of computational cost while maintaining the scientific performance of the global simulation for a specific



region. It is unclear if this conclusion would be robust for other regions of the globe where the incoming large-scale forcing may be more sensitive to the model resolution than the synoptically scale ARs that strongly influence CA's annual precipitation.

Nonetheless, despite this modest sensitivity we found that SCREAM has a satisfactory representation of AR events in both the "global proxy" configuration and the most economical RRM. The most significant issue found is a positive precipitation bias over topography, which is likely an issue in other regions of the globe in SCREAM that warrants further investigation.

*Code and data availability.* The code to run the simulations in addition to the model output used to generate the analysis in this paper can be found at https://doi.org/10.5281/zenodo.10836035. The PRISM data can be obtained at https://prism.oregonstate.edu, the SNOTEL data at https://nwcc-apps.sc.egov.usda.gov/imap/, the University of Arizona (UA) SWE data at https://nsidc.org/data/nsidc-0719/versions/1, and the ERA5 product at https://cds.climate.copernicus.eu/cdsapp#!/dataset/reanalysis-era5-single-levels?tab=overview.

*Author contributions.* Bogenschutz created each grid configuration, performed the simulations, and led the analysis. Zhang assisted in the creation of each grid configuration and participated in the analysis. Tang assisted in the creation of the original CA RRM grid. Cameron-Smith obtained funding for this work.

*Competing interests.* The authors declare that they have no competing interests.

*Acknowledgements.* This work is supported by the Lawrence Livermore National Laboratory (LLNL) LDRD projects [22-SI-008], "Climate Resilience for National Security". Work at LLNL was performed under the auspices of the U.S. DOE by Lawrence Livermore National Laboratory under contract DE-AC52-07NA27344. IM release: LLNL-JRNL-861973.



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



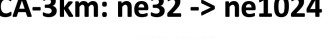

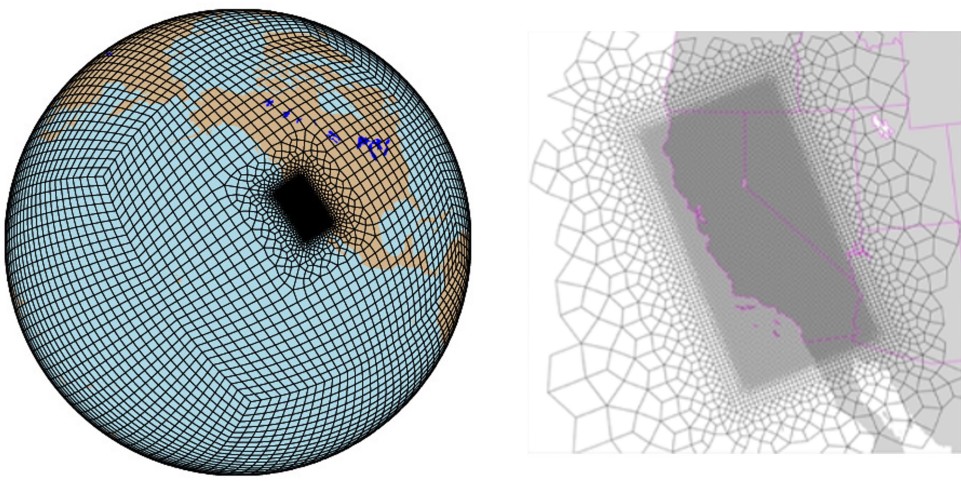

**Figure 1.** Depiction of the grid mesh used for the CA-3km CA RRM grid configuration this paper with ne1024 (~3 km) resolution used within the refined region and ne32 (~100 km) resolution outside of this region.

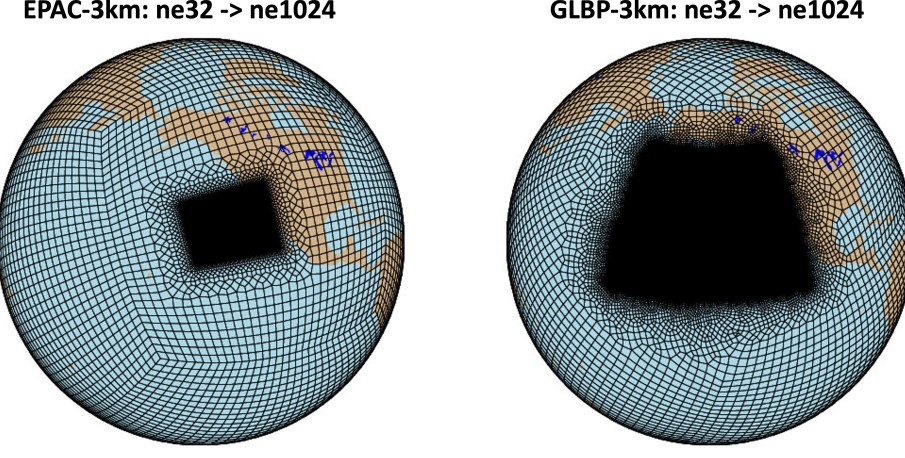

**Figure 2.** Depiction of the grid mesh used for the (left) EPAC-3km and (right) GLBP-3km grid configurations. Both configurations have ne1024 (∼3 km) resolution within the refined region and ne21 (∼100 km) resolution outside of this region.

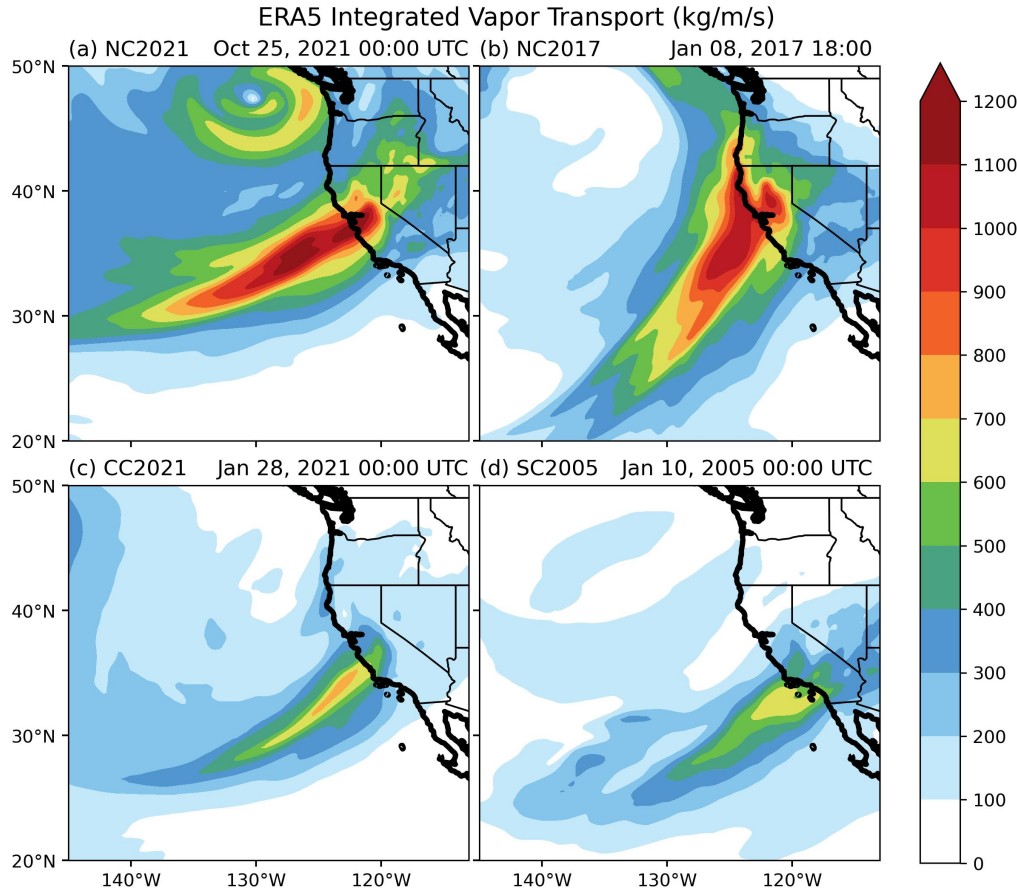

**Figure 3.** Integrated vapor transports as computed from ERA5 at the time peak intensity. Section 3.2 provides a meteorological description of each case, with case specifics summarized in table 2.



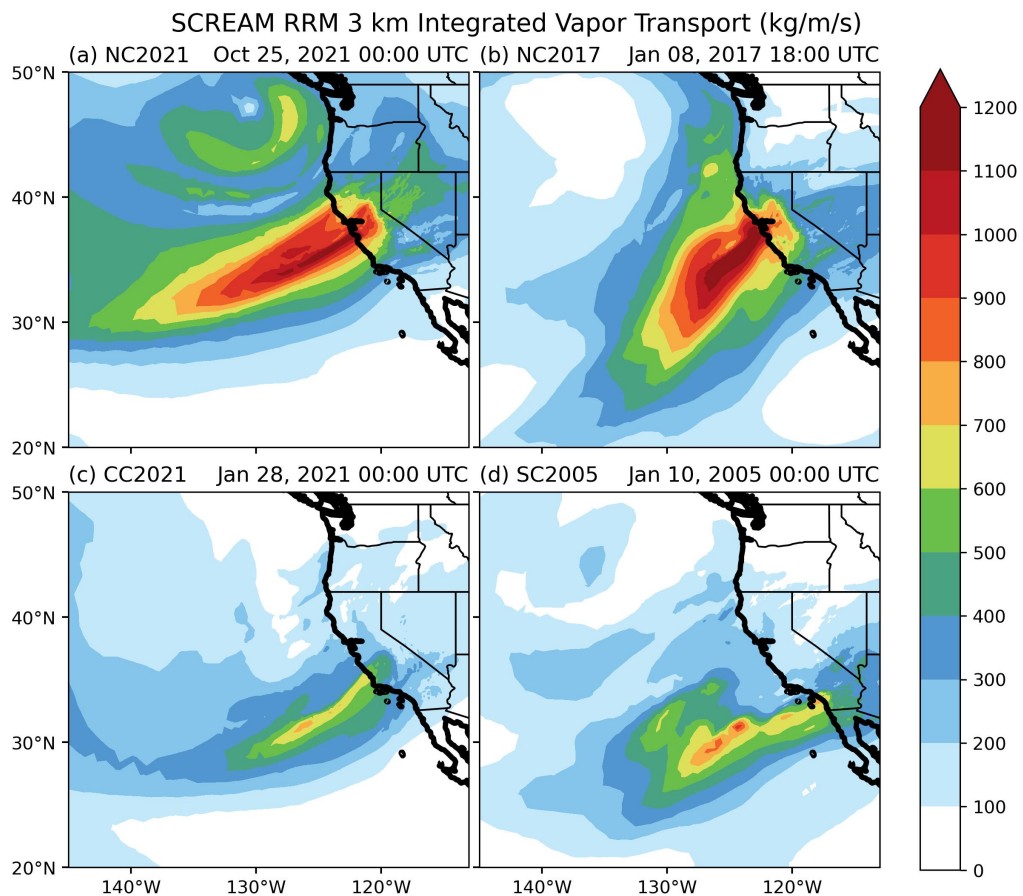

**Figure 4.** Same as Fig. 3 but for SCREAM RRM CA-3km simulations.



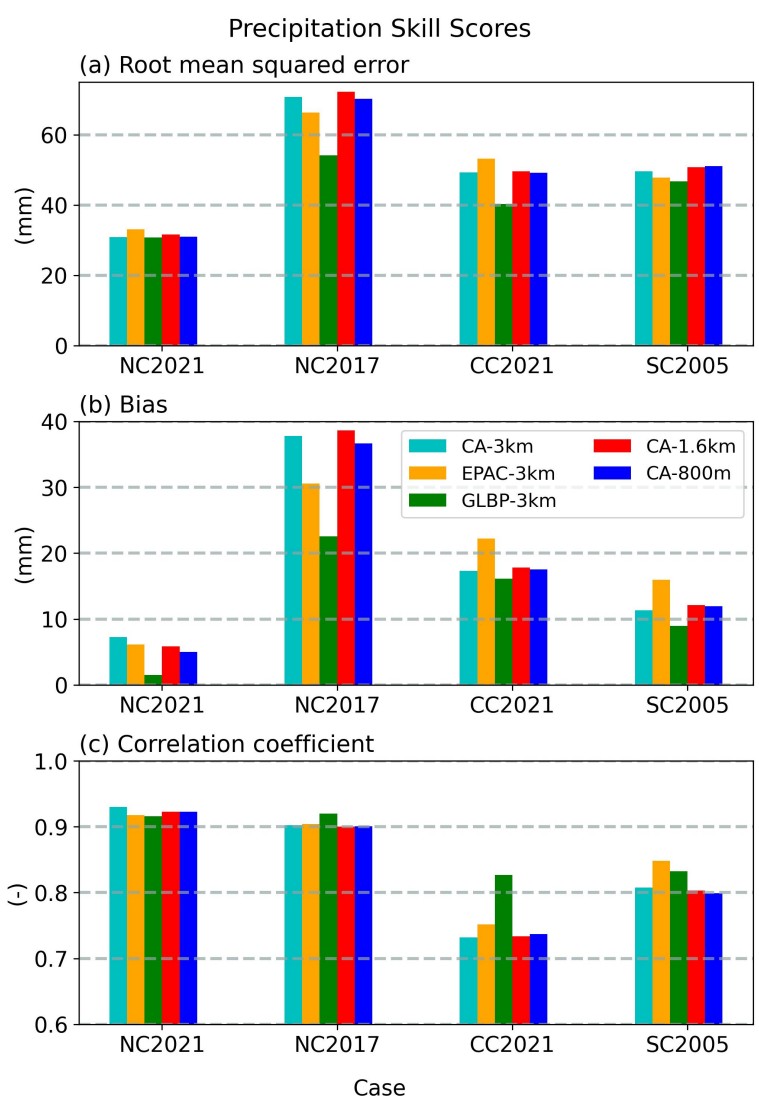

**Figure 5.** Skill scores for the event accumulated precipitation for each case and grid configuration as computed with respect to PRISM observations. Displayed is a) Root mean squared error, b) bias, and c) correlation coefficient.







**Figure 6.** Event accumulated precipitation for (top row) NC2021, (second row) NC2017, (third row) CC2021, and (bottom row) SC2005. Displayed are the (left column) PRISM observations, (second column) CA-3km RRM configuration, (third column) CA-800m configuration, and (right column) GLBP-3km grid configurations. Numerical value in the bottom left corner of each panel depicts the averaged value for CA. All data is plotted on the observation or model native grid.



**Figure 7.** Event accumulated precipitation bias computed relative to PRISM observations for (left column) CA-3km RRM, (middle column) CA-800m, and (right column) GLBP-3km for each case. Numerical value in the bottom left corner of each panel represents the averaged bias over CA. Model simulations are remapped to PRISM resolution to compute bias.

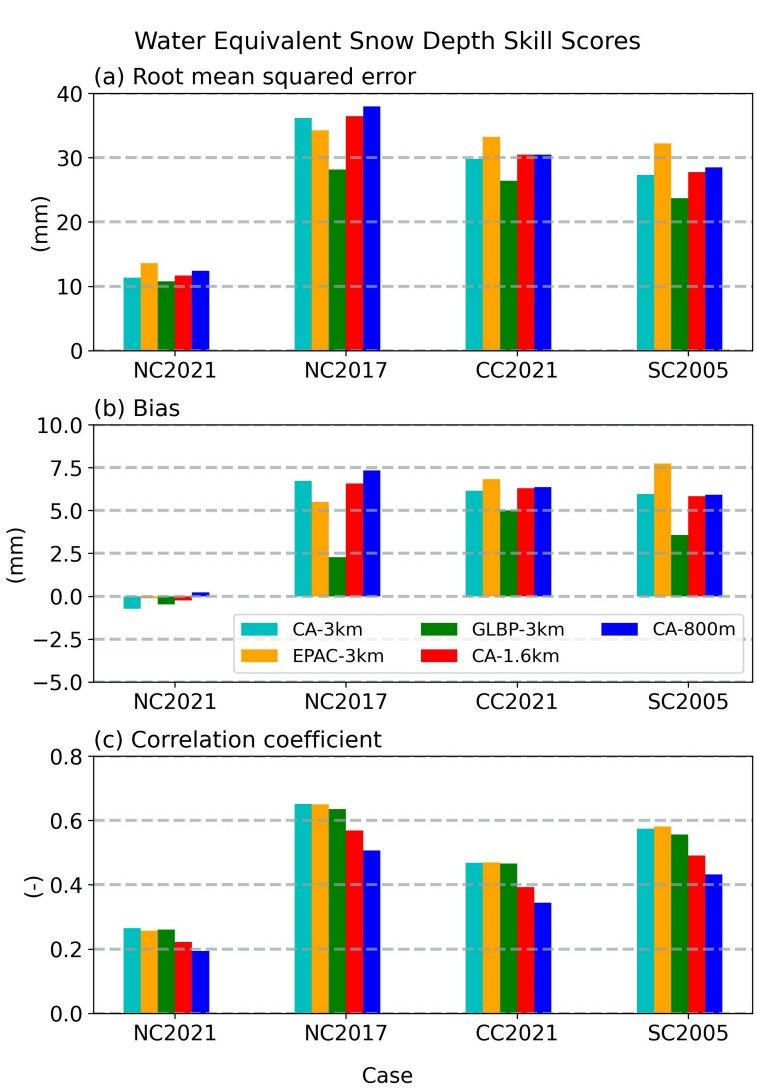

**Figure 8.** Same as Fig. 6 but for water equivalent snow depth computed to University of Arizona observational dataset.





**Figure 9.** Same as Fig. 6 but for water equivalent snow depth.



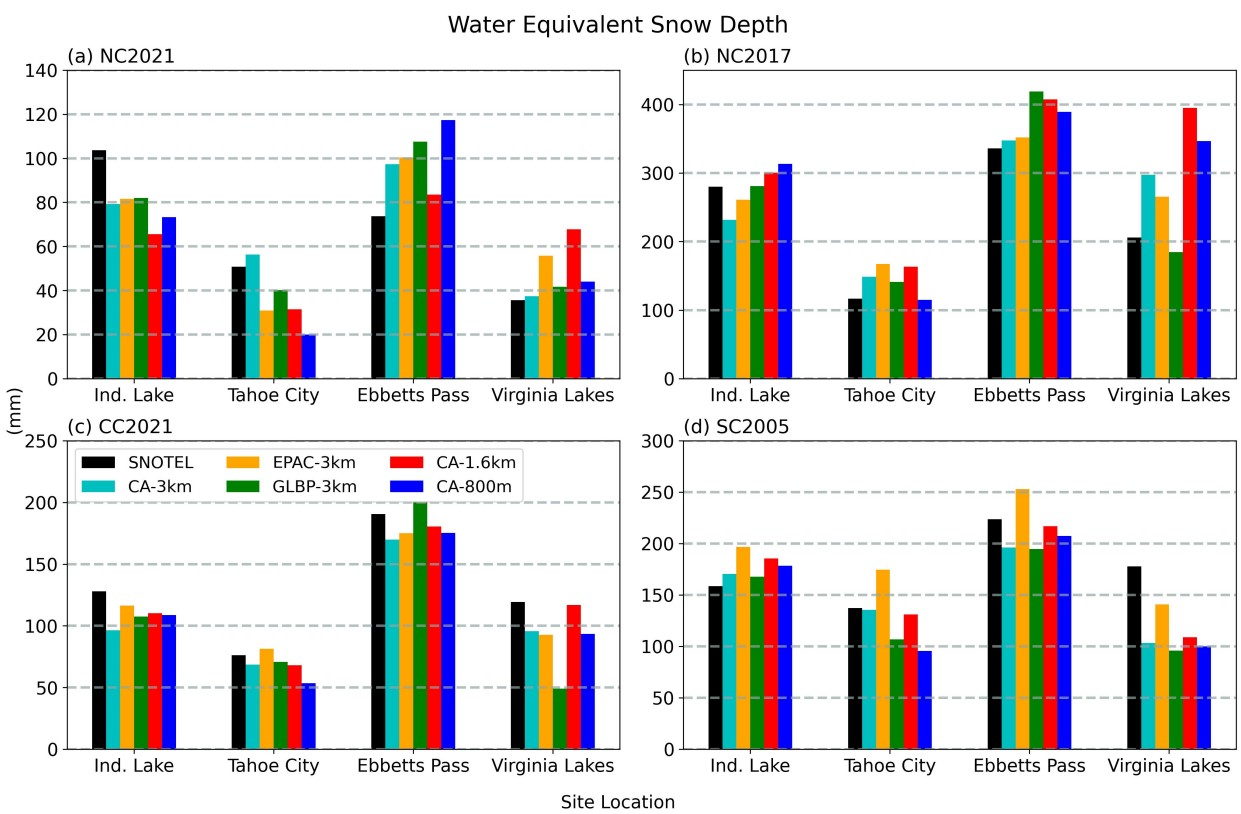

**Figure 10.** Event accumulated water equivalent snow depth for each case and grid configuration, compared to SNOTEL observations, at four different site locations. Elevation and location for each location can be found in Table 3. Data from model simulation is taken from the native grid.





**Figure 11.** Same as Fig. 5 but for 2 m temperature.



**Figure 12.** Same as Fig. 6 but for the event maximum integrated vapor transport with (left column) ERA5 depicted as the observational reference.





**Figure 13.** Same as Fig. 12 but for the event mean integrated vapor transport.

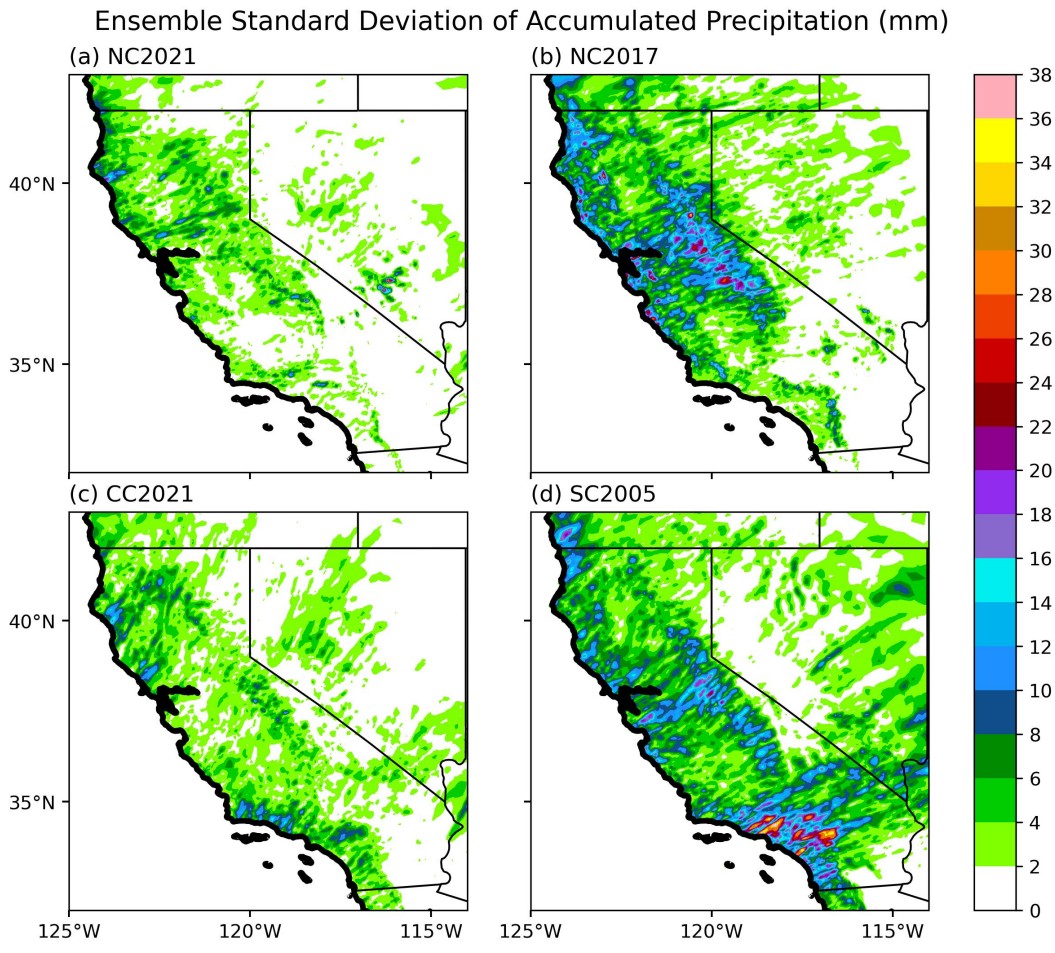

**Figure 14.** Standard deviation of event accumulated precipitation computed from a 10 member ensemble for each case for the CA-3km grid.



**Table 1.** Description of various RRM grids.

| Simulation ID | Effective resolution (km) | Number of physics columns | Number of dynamics columns | Physics time step (s) | Dynamics time step (s) |
|---|---|---|---|---|---|
| CA-3km | 100 to 3.25 | 67,872 | 152,714 | 75 | 9.375 |
| EPAC-3km | 100 to 3.25 | 103,536 | 232,958 | 75 | 9.375 |
| GLBP-3km | 100 to 3.25 | 1,251,844 | 2,816,651 | 75 | 9.375 |
| CA-1.6km | 100 to 1.6 | 175,8224 | 395,606 | 75 | 4.6875 |
| CA-800m | 100 to 0.79 | 587,904 | 1,322,786 | 75 | 2.34375 |

**Table 2.** Description of Atmospheric River Cases simulated by SCREAM RRM.

| Case ID | Initialization time | Simulation duration (hr) |
|---|---|---|
| NC2021 | 24-Oct-2021 00:00 UTC | 72 |
| NC2017 | 07-Jan-2017 00:00 UTC | 120 |
| CC2021 | 26-Jan-2021 12:00 UTC | 96 |
| SC2005 | 07-Jan-2005 00:00 UTC | 120 |

**Table 3.** Elevation and location information for the SNOTEL sites used for observational reference.

| Station | Elevation (m) | Latitude | Longitude |
|---|---|---|---|
| Independence Lake | 2541 | 39.43°N | 120.32°W |
| Tahoe City Cross | 2072 | 39.17°N | 120.15°W |
| Ebbetts Pass | 2640 | 38.55°N | 119.8°W |
| Virginia Lakes | 2877 | 38.06°N | 119.23°W |