# Peer review of "Atmospheric River Induced Precipitation in California as Simulated by the Regionally Refined Simple Convective Resolving E3SM Atmosphere Model (SCREAM) Version 0"

_EGUsphere, 2024_

## Author Comment (AC1)

Author response in red font, original editor/reviewer comments in black font.

Summary:

Bogenshutz et al. present four detailed case studies chosen to highlight different types and aspects of ARs impacting the California coast using the ultra-high regionally refined E3SM atmosphere model, SCREAM. The purpose of the study is to show SCREAM's potential utility simulating ARs for both climate and weather communities.  Their main findings, aside from the realism of AR representation, include (1) refining resolution beyond 3km has little impact for improving precipitation biases, but (2) expanding the spatial extent of the refinement has modest benefits.

Overall Comments:

This is a very nice paper. It is clear, well-written and organized, and absolutely fits within the GMD scope. I have a few comments I would like to see addressed before full publication.  Although these are relatively minor, they are important for clarity.  Please see specific comments below, but the most important ones are regarding SCREAM model description and method/domain for computing max/mean IVT.

Specific Comments:

Line 18 and elsewhere: Typically the words "atmospheric rivers" are not capitalized.

All instances of this have been corrected.

Line 89: Out of curiosity, when you moved to using SHOC from a simplified PBL, was there anything that needed to be re-tuned?

SCREAM has always used SHOC and does not have the option to run a simplified PBL scheme.  SHOC was not tuned in any manner for this study.

Figure 2: Did you mean ne32 and not ne21?

Yes, this has been corrected.  Thanks for the catch!

SCREAM description and References:  There are missing citations in the reference list, I definitely noticed both Caldwell et al 2021 and 2019 missing as I searched for these to get more information on SCREAM. Also, I recommend adding a bit more to the

description part of the paper for ease rather than digging into the cited Caldwell papers, i.e. vertical resolution, features, biases, improvements not already mentioned, etc.

Thank you for pointing out the inconsistency related to the references.  The mentioned references have been added to the list (in addition to others that were found missing).

Good suggestion.  We added the following paragraph:

"SCREAM features 128 vertical layers, compared to 72 in E3SM, despite having a lower model top (40 km vs. 60 km). This gives SCREAM nearly twice the vertical resolution of E3SM in most layers, with particularly enhanced resolution in the lower troposphere. The improved resolution in the lower troposphere has been found to enhance the representation of marine stratocumulus, which is crucial for accurately depicting the California coastal climate (Bogenschutz et al., 2021, 2023; Lee et al. 2021, 2022). Although SCREAM offers improvements in many simulated aspects over the lower resolution E3SM model (as detailed in section \ref{intro}), it still faces challenges in representing shallow cumulus convection (Bogenschutz et al., 2023) and struggles with aggregating convection in the deep tropics compared to observations (Caldwell et al., 2021)."

AR cases (these are more curiosity questions than comments on the paper itself):  There are all great choices and span important and different types of ARs and dynamics.  I am curious, however, on how SCREAM-RRR would perform for 1) a weak Cat 1 storm, or 2)  different regions , or 3) wet vs windy "flavors"? Perhaps these are already planned activities, and I am not suggesting these be done for this paper. (I understand the intense computational and person-resources involved in these activities).  Some questions to consider for future experiments (or answer if you have already done some of this):  For a weak category 1 AR, or a "windy" AR,  are there components of physics, dycore, and/or resolution that do better with an overwet environment vs not?  Are biases different across the varied AR dynamics, i.e. Pineapple Express vs North Atlantic warm conveyor belt lift? (Has this been looked at)? Are there plans to look at moisture transports such as ARs within monsoonal flow which can be found in the E.Pacific?

The reviewer brings up many great points and suggestions.  We agree that the cases presented in this paper, while diverse, certainly are not exhaustive and additional studies/cases are definitely warranted.  While none of this work is currently planned, as the SCREAM project operates on very limited resources, I imagine that many of these topics will/can be explored once SCREAM becomes the default atmosphere model of

E3SM (EAMxx) and FTE increases.  In addition, EAMxx will be written in C++/Kokkos to allow efficiently throughput on GPUs, which will facilitate more experimentation.

We added the following statement to the end of the paper: "Finally, future work should focus on the simulation ARs in SCREAM in regions outside of California, either through global simulation or RRM, to encompass a broader range of ARs and meteorological conditions."

Line 148 and elsewhere: The acronym "AR" only needs to be defined once.

All redundant definition instances have been removed.

Line 253:  Why not add polygons to one of your maps with the locations of each of your evaluation points? This would be helpful visualizing where the skill is being evaluated. (Or maybe an inset with a zoomed map)?

Great idea.  We added an additional figure to denote various frequently mentioned geographic features in CA (as suggested by reviewer 2) and we added the locations of the SNOTEL sites to this map.  We added the following text:

"For convenience, we have included Figure~\ref{CAref}, which shows the locations of various California geographic features, various points of interest, and evaluation sites that are frequently mentioned throughout this paper when discussing the cases and results."

Line 287 and Figure 5c: Pearson pattern correlation?

Yes, clarified in both the text and figure caption.

Line 327: Comparing SC2005 reanalysis and SCREAM, the character of the IVT plume and footprint are very different. One might argue that SCREAM really doesn't capture the AR structure well here, especially in the ocean. Ideas as to why?

Yes, we agree with the reviewer, and this is noted in the text. Unfortunately, we don't have any clear explanation as to why SCREAM struggles more with this case than others. We tried several offline experiments to improve the representation in the CA-3km grid, including initializing the model closer to AR landfall, but these did not improve the precipitation accumulation pattern. Additionally, all ensemble members of our experiments produced the same general result. We conducted experiments where the coarse outer grid was weakly nudged to ERA5, but the results were not significantly different. However, we did not experiment with stronger nudging. We believe that

additional similar cases would need to be examined to determine if this problem persists under similar synoptic conditions.

Lines 359-360:  I understand the tension between the improvements via resolution and mesh area, but the other degree of freedom here is E3SM vs CESM when consulting with the results from the Rhoades paper. I think(?) that the vertical levels are different between these two modelling frameworks? How different is SCREAM vertical structure from the Rhoades study, and do you think this may matter regarding convection schemes?

The reviewer brings up a good point that the discussion on these lines is agnostic to the model choice (i.e. E3SM vs CESM).  While the vertical layers are different between these two models, we hypothesize that the choice of vertical grid would not impact our conclusions.  Currently, the SCREAM development team is doing vertical resolution sensitivity experiments (to be documented in an upcoming paper) and we are finding that our model is ridiculously robust to vertical resolution (which is a good thing; but also a bad thing that pre-existing biases don't go away merely by changing the resolution).  I would suspect that there could be a larger sensitivity to the choice of physical parameterizations (especially microphysics and to a lesser degree turbulence).

At the end of this section we add the following text:  "However, we acknowledge that comparing our results to other works, such as Rhoades et al. (2020) and Huang et al. (2020), involves being agnostic to model differences that may not justify a direct comparison, such as variations in physics, dynamical core, and vertical levels."

Line 401: Did you mean Figure 10 (and not 3)?

Yes, thank you for point this out.  This has been fixed.

Figure 10:  I find it interesting that the EPAC 3km case seems to overdo the snow compared to the CA 3km in most cases, i.e. most notably at Virginia Lakes for NC2021 and everywhere SC2005. Thoughts?

The reviewer brings up an interesting point.  There could be no apparent reason found for the NC2021 case, though this increased snow amount is comparatively quite small compared to the other cases given the smaller scale on the y-axis.

For the SC2005 case, the EPAC configuration does have increased amounts for all sites. Upon investigation, it was found that the IVT values produced by this configuration are slightly higher than the other configurations, which led to the increased amounts of precipitation that are most apparent in the southern and central Sierra.  We note that

other than this slight increase in precipitation amounts, this configuration looks characteristically similar to the other configurations for the SC2005.

We have added the text: "It is interesting to note that the EPAC-3km configuration generates more snow at all sites compared to the other configurations. This aligns with the slightly higher positive bias observed in statewide precipitation and snow (Figs.~\ref{prect_skill_bar} and~\ref{snowh_skillscore_bar}, respectively) for this configuration."

Line 428, Figures 12/13, Lines 440-447:  For the event maximum/mean IVT, did you use find, at each grid point in your RR domain, the maximum/mean values and plot this? Or just across the plotting extent in Fig 12/13? For SC2005 where SCREAM's IVT core could arguably be over the ocean (based on Figure 4), this might be relevant. Are some of SCREAMs biases due to AR characterization and misplacement of the core, and not really just the fact that the plume hitting the coast is more narrow? For clarity, please add a sentence or two clarifying IVT max/mean methodology.

The max/min values represents the respective values for each grid column for model and analysis.  This has been clarified in the text.

---

## Author Comment (AC2)

Author response in red font, original editor/reviewer comments in black font.

The study examines precipitation, snowfall, and surface temperature characteristics associated with 4 atmospheric river (AR) cases impacting California as simulated in the Simple Cloud Resolving E3SM (Energy Exascale Earth System Model) Atmosphere Model (SCREAM) with selected regionally refined grid mesh (RRM) configurations.  Sensitivity of the results to the RRM horizontal resolution (3.25 km, 1.6 km, and 800 m) and upstream extent of the RRM domain is evaluated against both high-resolution gridded data sets and single-station time series.  Generally, SCREAM is able to capture the fine-scale regional distributions of AR-related precipitation over California.  However, the model systematically overestimates precipitation over upwind and higher-elevation areas, and underestimates precipitation on leeward sides of mountain ranges.  SCREAM also tends to overestimate storm-total snowfall and mean surface temperatures during the AR events.  Overall, it is found that precipitation and temperature distributions exhibit only modest sensitivity to RRM resolution.  Although somewhat greater improvement is seen for the largest RRM domain extent due to better representation of large-scale meteorological patterns that guide the ARs, the authors assert that the required RRM domain expansion no longer achieves the intended computational cost savings.

I found the modeling approach and scientific assessment to be both clear and informative.  The detail of the model evaluation was appropriate given the fairly large number of simulations conducted and AR cases examined.

A key finding of the paper, in my view, is the relative insensitivity of storm-total precipitation to model horizontal grid resolution in the range from 3.25 km to 800 m.  This finding also highlights the current  horizontal resolution limit of evaluation data sets (4 km for daily surface precipitation, and 25 km for ERA5 atmospheric fields).  At such fine scales, the need would arise to start evaluating models against single meteorological stations, as the authors have done here.

Findings related to the RRM domain extent are also novel.  The authors note that SCREAM is initialized with ERA5 but allowed to evolve freely thereafter.  In my opinion, initializing the model 24 hours (or 18 or 30 hours, based on sensitivity tests) is perhaps a "low bar" to assess model performance, though I suppose longer lead times might result in a drift of the large-scale meteorological solution such that evaluation of AR impacts could become complicated.  I view the paper's findings mostly as an examination of AR impacts from simulations in which the large-scale meteorological patterns are quasi-prescribed, given the short lead time.

The authors thank the reviewer for their thoughtful and thorough review. We have revised the manuscript accordingly. Please see our responses for each relevant point below.

**Specific comments:**

L26-28: Rutz et al. (2014, https://doi.org/10.1175/MWR-D-13-00168.1) displays one of the earliest and nicest maps of AR contributions to total cool-season precipitation (their Fig. 8). Consider adding this reference here.

Done.

L49-51: Can a reference to these modeling improvement be added here?

The Caldwell et al. (2021) is the appropriate reference here; which was cited in the preceding and following sentences.

L94-97: "SCREAMv0 used a prescribed-aerosol version of E3SM's modal aerosol model, however in this work we use a version of SCREAM that prescribes both cloud-condensation nuclei number and aerosol radiative properties from an E3SMv2 simulation. This is known as Simple Prescribed Aerosol (SPA) and will be incorporated into SCREAMv1." Do the authors think that this simplification, especially the prescribed CCN, could potentially be contributing to the "eagerness" of SCREAM to overestimate precipitation on the upwind side of mountain ranges?

This is a good point. While it is unclear to us if this could be a contributing factor, this is something that could be explored via sensitivity studies (as alluded to in the conclusions) with this configuration.

Figs. 2 and 3: In future depictions of RRMs, it is recommended to recreate the plot formatting like the right panel in Fig. 1 so that land masses and coastlines are visible.

We thank the reviewer for this suggestion and we agree this would be better to do.

L159: "most locations" is too vague

Agreed. Changed to "in addition to widespread reports of damaging winds in excess of 27 m/s that resulted in extensive power outages"

L163-164: ERA5 and IVT should be defined here, as it's their first usage.

Done.

L169-170:  D4 drought classification should be referenced.

Done.

L214-216:  Three references on the large-scale meteorological patterns that precede west coast (of U.S.) AR landfalls:

References have been cited.

——

Benedict, J. J., Clement, A. C., & Medeiros, B. (2019). Atmospheric blocking and other large-scale precursor patterns of landfalling atmospheric rivers in the North Pacific: A CESM2 study. Journal of Geophysical Research: Atmospheres, 124, 11,330–11,353. https://doi.org/10.1029/2019JD030790

Zhou, Y., & Kim, H. (2019). Impact of Distinct Origin Locations on the Life Cycles of Landfalling Atmospheric Rivers Over the U.S. West Coast. Journal of Geophysical Research: Atmospheres, 124, 11,897–11,909. https://doi.org/10.1029/2019JD031218

Carrera, M. L., Higgins, R. W., & Kousky, V. E. (2004). Downstream weather impacts associated with atmospheric blocking over the northeast Pacific. Journal of Climate, 17(24), 4823–4839. https://doi.org/10.1175/jcli-3237.1

——

L248-249:  SNOTEL and Cooperative Observer Program should be referenced.

Done.

L273-274:  It would be helpful to clarify how "bias" is calculated, as well as what spatial domain is used in the calculations for Fig. 5.  The author's use of "statewide" in L276 leads me to believe that the entire state of California is being used as the spatial domain for evaluation, but this should be more clearly stated.

Yes, this desperately needed to be clarified.  We have added the following passage: "We note that all skill scores presented are statewide, using only the model columns and points for SCREAM and PRISM that fall within California. These are determined by a mask file generated from a California shapefile."

A map containing locations (mountain ranges, counties) highlighted in the text should be added to help readers not familiar with California geography.

Great suggestion.  We have added Figure three which includes the geographic points of interest in addition to the locations of the SNOTEL evaluation sites (suggested by reviewer 1).  We have added the following text to the document:

"For convenience, we have included Figure~\ref{CAref}, which shows the locations of various California geographic features, various points of interest, and evaluation sites that are frequently mentioned throughout this paper when discussing the cases and results."

In Figs. 12 and 13, what temporal resolutions are used for SCREAM and ERA5 to evaluate maximum IVT?  This should be noted in the L428 paragraph.

We have added the sentence: We note that the temporal resolution of ERA5 is coarser (3 hours) compared to the 1-hour resolution used in the SCREAM analysis.

L486-494:  It would greatly benefit the paper to add references to specific figures when summarizing the key findings.

Good idea, done.

L511:  At the end of this paragraph, it might be good to add a sentence noting that model evaluation at sub-kilometer scales is challenged by the availability of observations needed to evaluate the simulations… though I suppose a comparison to a single station would be appropriate in some cases.

We have added the sentence: "However, we acknowledge the added challenge of evaluating sub-kilometer scale models due to the lack of available high-resolution observations."

**Technical Corrections:**

All technical corrections have been addressed.  Thank you for the careful check!

L2:  E3SM should be defined.

L42:  "generation…have" —> "generation…has"

L62:  "Winter Hydroclimate" need not be capitalized.

L73-74:  "resolution is increased to 1.6 km":  It would be better to state something like "resolution is increased from 3 km to 1.6 km…"

L77:  "5 km though, that work" —> "5 km, though that work"

The Fig. 1 caption seems to be missing one or more words.

Fig. 2 caption:  Should be "ne32", not "ne21".

L135:  Missing period at end of sentence.

L171:  ;  —> ,

 L215:  "set up" (verb) —> "setup" (noun)

 L232:  letting —> allowing

 L281-283:  Remove "Though", add comma after "scores", and change semicolon to comma.  Also, "grids and there can be implications" —> "grids.  There can be implications…"

 L310:  ; —> ,

 L388:  models —> model's

 L401:  Should be Fig. 10, not Fig. 3.

 L402:  "though" —> "though they"

 L403:  Lakes —> Lake

 L405:  ; —> ,

 L425:  ; —> ,

 L455:  ; —> ,

 L495:  "The aforementioned positive precipitation bias seen in our simulations of 10 to 33%, is far less…" — please change to:  "The aforementioned positive precipitation bias of 10-33% seen in our simulations is far less…"

 L498:  Remove semicolon

 L505:  "SCREAM's resolution" —> "SCREAM's muted resolution"…?